# The Importance of a Distance between the Lines Encircling Pulmonary Veins in Atrial Fibrillation Ablation on First-Pass Isolation Ratio and Clinical Outcomes

**DOI:** 10.3390/ijerph20075250

**Published:** 2023-03-24

**Authors:** Krystian Krzyżanowski, Marek Kiliszek, Beata Uziębło-Życzkowska, Magdalena Smalc-Stasiak, Aleksandra Winkler, Paweł Krzesiński

**Affiliations:** Department of Cardiology and Internal Diseases, Military Institute of Medicine, 04-141 Warsaw, Poland

**Keywords:** ablation, atrial fibrillation, pulmonary vein isolation, posterior wall, wide antral circumferential ablation

## Abstract

Introduction: How wide the encircling line is made may influence the outcomes of pulmonary vein isolation (PVI). In the present study we hypothesised that the distance between the lines encircling the pulmonary veins may correspond with the extent of wide antral circumferential ablation (WACA). The aim of the study was to assess the impact of the distance and the area between the lines on the posterior wall of the left atrium on first-pass isolation rate and 12-month freedom from atrial arrhythmia in patients undergoing PVI ablation. Methods and results: One hundred sixteen patients underwent circumferential ablation index (AI)-guided PVI. The distance between the encircling ablation lines was measured off-line between the uppermost points (right and left) and the lowest points and as the area between the encircling lines on the posterior wall. The first-pass isolation rate and 12-month freedom from atrial arrhythmia were 59% and 73%, respectively. Distance between the encircling lines measured linearly or as the area of the posterior wall, assessed as direct values or indexed to left atrial dimensions, did not differ between patients with and without first-pass isolation or between patients with and without recurrences of atrial arrhythmia. Conclusions: The distance between the ablation lines did not influence the rate of first-pass isolation and arrhythmia recurrence in the long-term follow-up after PVI procedures incorporating the ablation index protocol.

## 1. Introduction

Pulmonary vein (PV) isolation is a cornerstone of contemporary ablation procedures in patients with atrial fibrillation (AF). Since its introduction, the ablation index (AI), a formula incorporating power, contact force, and stability of the catheter, has allowed for the formation of more efficient and durable ablation lines encircling the pulmonary veins. The use of AI was found to increase the first-pass isolation of the PV, reduce the incidence of acute and delayed PV reconnections, and improve arrhythmia-free survival after a single procedure [1,2]. The advantage of AI-guided ablation results from providing contiguous lesions of predictable and stable size. However, first-pass isolation is not achieved in every patient, and recurrences of atrial fibrillation still occur. Moreover, some patients require additional ablation in the carina region after completion of the encircling line to achieve electrical isolation of the pulmonary veins. It is acknowledged that how wide the antral ablation lines have been made plays an important role in the efficacy and safety of pulmonary vein isolation (PVI). Previous studies, before the introduction of the ablation index, showed that both the first-pass isolation rate and freedom from AF depended on the distance between the ablation line and the ostia of the PV [3,4]. Since a considerable number of PVI procedures are performed using three-dimensional electroanatomical system and fluoroscopy only, it is often difficult to precisely delineate the ostia of the PV.

We hypothesised that the measurements of the distance between the encircling ablation lines, right and left, may be a surrogate for the extent of wide antral ablation of the PV. Therefore the purpose of this study was to evaluate the impact of the distance between the ablation lines, measured linearly or as the mean area, on the first-pass isolation rate and freedom from arrhythmia in long-term follow-up.

## 2. Methods

This retrospective, single-centre, observational study was conducted on 116 consecutive patients undergoing PVI radiofrequency ablation between August 2017 and June 2019 due to paroxysmal, persistent, or long-term persistent AF. All patients were qualified for PVI according to current guidelines [5]. The patients who required additional ablation in the left atrium outside the antral region of the pulmonary veins were excluded from the study. The ablation procedure was performed under conscious sedation. The left atrium was accessed through a double transseptal puncture guided by continuous pressure measurements and fluoroscopy. Unfractionated heparin was administered in boluses directly after the transseptal puncture to maintain activated clotting time between 300 and 350 s. A circumferential mapping catheter was used for mapping (Lasso, Biosense Webster, Irvine, CA, USA). Radiofrequency ablation was delivered using an irrigated contact force (CF) catheter (Navistar Thermocool Smart Touch, Biosense Webster, Irvine, CA, USA). Navigation of the catheters was based on fluoroscopy and on the electroanatomical CARTO 3 system (Biosense Webster, Yokneam, Israel). The ipsilateral veins were isolated in a single circle. The ablation index (AI) settings were as follows: the catheter stability range of motion was 3 mm, the catheter stability time was more than 3 s, and the CF was greater than 3 g more than 25% of the time. The power limit was 35 to 40 W. The AI threshold for the anterior wall was 500 and 380 for the roof and the posterior wall, respectively. The radiofrequency delivery was stopped at the posterior wall at an AI > 300 in cases of severe chest pain. The target intercession distance was 4–5 mm with maximal interlesion distance of 6 mm. The endpoint of the procedure was the isolation of all PVs. If the PVs were not isolated after completing the encircling lines, additional ablation was performed within the line or in the carina region if necessary. PVI was evaluated during a 20-min waiting period after successful isolation.

### 2.1. Follow-Up

The patients were scheduled for two follow-up visits after 6 and 12 months. On each follow-up visit, the patients were asked for any electrocardiographic documentation. In patients who were symptomatic and presented ECG with atrial fibrillation, atrial flutter, or atrial tachycardia after the 3-month blanking period, recurrence was confirmed. The rest of the patients had an ECG performed on each follow-up visit: atrial fibrillation, atrial flutter, and atrial tachycardia were considered recurrence. Patients without the electrocardiographic evidence of atrial arrhythmia were scheduled for a 7-day Holter monitoring at each follow-up visit. Any atrial tachycardia lasting more than 30 s in the 7-day Holter monitoring was defined as a recurrence.

### 2.2. Measurements of the Distance between the Encircling Ablation Lines

The distance between the encircling ablation lines was measured off-line by an experienced electrophysiologist blinded to the clinical outcome. In each right and left ablation line, the uppermost point and the lowest point were identified. The upper distance and the lower distance were measured between the uppermost points (right and left) and the lowest points (right and left), respectively (Figure 1 and Figure 2). Then, the area on the posterior wall between the right and left ablation lines and the upper and lower borders (identified previously) was selected and measured (Figure 3). The area was analysed as a raw value or as a quotient of the area measured divided by the left atrial diameter or as a quotient of the area measured divided by the left atrial area.

### 2.3. Statistical Analysis

Continuous variables are presented as the mean (standard deviation, SD). Categorical variables are presented as frequencies. Normality of distribution was tested with the Shapiro–Wilk test. Comparisons of the parameters were performed with Student’s t-test (normal distribution, continuous variables) or the Mann–Whitney U-test (continuous variables, distribution other than normal). A two-side *p*-value < 0.05 was considered statistically significant. Univariate and multivariate logistic regression analysis was performed to assess the relationships of recurrence of arrhythmia and the distance between the ablation lines with other variables. Variables selected to be tested in the multivariate analysis were the parameters of the distance between ablation lines, age, and those with a *p*-value of <0.2 in the univariate analysis. All calculations were performed with Statistica software, version 12 (Statsoft Inc., Tulsa, OK, USA).

## 3. Results

Seventy-seven patients with paroxysmal, 27 with persistent and 12 with long-term persistent AF were included in the study. Demographic data and clinical characteristics are provided in Table 1. First-pass isolation of both the right and left PVs occurred in 68 (59%) patients. Forty-eight patients required additional ablation within the primary line or in the carina region. There were complications in four patients: two femoral pseudoaneurysms, one femoral atriovenous fistula, and one femoral pseudoaneurysm resulting in stroke and subsequent death.

During the 12 months follow-up period, 85 (73%) patients were free from arrhythmia recurrences. The mean distance between ablation lines was 34.8 ± 8.5 mm with a range of 17.3–57.3 mm (the uppermost ablation points) and 48.7 ± 10.2 mm with a range of 17.2–74.4 mm (the lower ablation points). The mean area on the posterior wall between the right and left ablation lines and the upper and lower borders was 13.4 ± 4.2 cm^2^ with a range 6.5–26.4 cm^2^. The distance between the ablation lines, measured linearly, as the mean area, or as the mean area indexed to the left atrial dimension or the left atrial area (measured with echocardiography), did not differ between patients with and without first-pass isolation (Table 2) or between patients with and without recurrence of arrhythmia (Table 3). The impact of clinical factors on the arrhythmia recurrence rate is presented in Table 4. The lower ejection fraction was the only predictor of arrhythmia recurrence. In the multivariate analysis, none of the analysed parameters was significantly correlated with arrhythmia recurrence (Appendix A).

## 4. Discussion

In recent years, pulmonary vein isolation ablation has evolved from ostial segmental isolation of each pulmonary vein to wide antral circumferential ablation (WACA) of the ipsilateral veins [6,7]. The ablation of a larger area around the PV was found to be more effective in terms of recurrences of arrhythmia and, at the same time, resulted in fewer cases of PV stenosis [7]. The theoretical basis for the superiority of a more extensive approach is that not only the pulmonary veins are involved in the pathophysiology of atrial fibrillation but also the atrial myocardium surrounding the pulmonary veins. It has been observed that arrhythmogenic foci, fibrotic area, and ganglionate plexi are located predominantly around the pulmonary veins and on the posterior wall of the left atrium [8,9,10]. The same theory led to the conception of routine posterior wall isolation for atrial fibrillation ablation, but randomised studies have yielded conflicting results [11,12,13]. Therefore, it is of great importance to find the optimal width for antral circumferential ablation around the pulmonary veins.

The definition of WACA is not precise. However, it is accepted that the ablation should be performed several millimetres away from the ostium of PV more than 0.5–1.0 cm on the anterior wall and more than 1.0–1.5 cm on the posterior wall. In the anterior aspect of the left superior PV it is often necessary to ablate within 0.5 mm from the ostium [6,7,14].

The degree of WACA may differ, and its impact on direct PVI efficacy and long-term freedom from arrhythmia have been studied previously. Lin et al. showed that the only predictor of the requirement for additional ablation in the carina region after the first round of circumferential isolation was the distance between the ablation line and the ostia of the PV [3]. The longer the distance was, the more frequent was the necessity for ablation in the carina region. However, a longer distance between ablation points of the primary circle and the carina region predicted better outcomes of the procedure in the long-term follow-up. This outcome is in agreement with the study of Mulder et al., who investigated the prognostic implications of gap-related and carina-related persistent conduction during pulmonary vein isolation [15]. They showed that although substantial numbers of patients required additional ablation in the carina region (right or left), it did not predict the recurrence of AF. Carina-related persistent conduction correlated with larger WACA circumference, indicating greater distance of the ablation line to the PV ostia. Kiuchi et al. examined the importance of the size of the isolated area of the left atrium after PVI for paroxysmal atrial fibrillation [4]. The isolation surface area was defined as the ratio of the isolated area between the ostia of the veins and the ablation lines on both sides to the total sum of the isolated surface area (as above) and the area of the posterior wall (between the ablation lines). A large ratio of isolation surface area (wide ablation) was positively correlated with freedom from atrial arrhythmia. Ablation was most effective in the group with a ratio of isolation surface area ≥70% and non-isolated posterior wall area of 9.5 cm^2^ with all 22 patients free from atrial arrhythmia. The mean area of the non-isolated posterior wall in the whole group was 13.9 cm^2^, similar to the findings of the present study.

The above-mentioned studies did not incorporate the ablation index protocol, which enables formation of more efficient transmural lesions. It is probable that adopting this technique allows for performing PVI less dependably from the specific site of ablation. However, in the study by Yoshimura et al., the prognostic value of the size of the non-isolated left atrial posterior wall was tested in 132 patients with non-paroxysmal AF who underwent ablation index-guided pulmonary vein isolation [16]. In the group with recurrence of atrial arrhythmia, the non-isolated posterior wall area was significantly greater than in patients without recurrences: 10.9 ± 3.5 cm^2^ and 8.2 ± 3.2 cm^2^, respectively, and 8.6 ± 3.3 cm^2^ for the whole group. The same was true for the ratio of the non-isolated posterior wall area to the left atrial surface—the lower the ratio, the better the outcome of the ablation. However, the size of the isolated area of the left atrium (the left atrial surface between the ostia of the PV and the ablation circles) did not predict freedom from atrial arrhythmia recurrence.

In the studies cited above, delineation of the PV ostia was augmented by integration of computed tomography or magnetic resonance imaging with models of the left atrium acquired by electroanatomical mapping systems or PV venography. It is not uncommon to perform PVI procedures using three-dimensional electroanatomical system and fluoroscopy only. Then, the exact identification of PV ostia may be difficult and variable between different physicians. In the present study, the distance between right and left ablation lines on the posterior wall, measured as linear distance or the surface area, was used as a surrogate for the distance of the ablation lines from the PV ostia. The measurements were analysed directly and after indexing them to left atrial dimensions. This approach allows for repetitive approximation of how wide the circumferential antral isolation is devoid of inter- and intra-observer variability. It can also be easily adopted in clinical settings since the measurement of the distance and the area between ablation lines on the posterior wall is not as difficult as delineation of the PV ostia. The mean value of the non-isolated posterior wall area in the whole group in the present study was similar to the findings of Kiuchi et al. [4] and greater than in the study of Yoshimura et al. [16]; the latter could have been influenced by ethnic differences.

### Limitations

In the present study, both the rate of first-pass isolation and long-term clinical outcomes of the ablation procedures were not associated with the width of antral isolation, differing from the findings of the previous two studies [4,16]. Although the range of the distance and the area between ablation lines were considerable (even if indexed to left atrial dimensions), it is possible that the single-centre single-operator study lacked the necessary variability. Additionally, more stringent follow-up methods (longer ECG monitoring or implantable loop recorders) could affect the results. Since the collection of this study′s data, trials showing better outcomes of ablation strategies beyond pulmonary vein isolation in patients with persistent and long-term persistent AF have been performed [17]. In the present study, only PVI was performed in these patients. Furthermore, the study group was heterogeneous, consisting of patients with paroxysmal, persistent, and long-term persistent AF. Further multi-centre studies are required to investigate the optimal degree of WACA.

## 5. Conclusions

The distance between the ablation lines on the left atrial posterior wall, measured linearly or as the mean area, did not influence the rate of first-pass isolation or arrhythmia recurrence in the long-term follow-up after PVI procedures incorporating the ablation index protocol.

## Figures and Tables

**Figure 1 ijerph-20-05250-f001:**
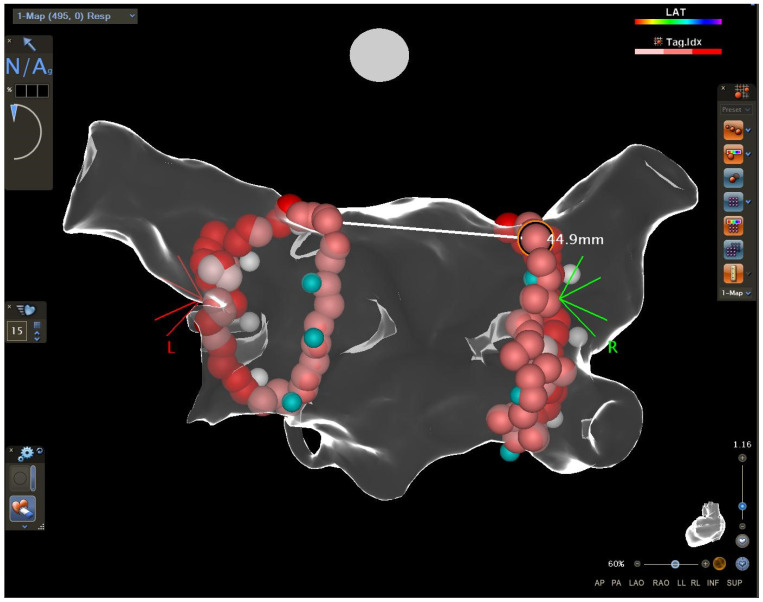
The measurement of the upper distance between the uppermost points.

**Figure 2 ijerph-20-05250-f002:**
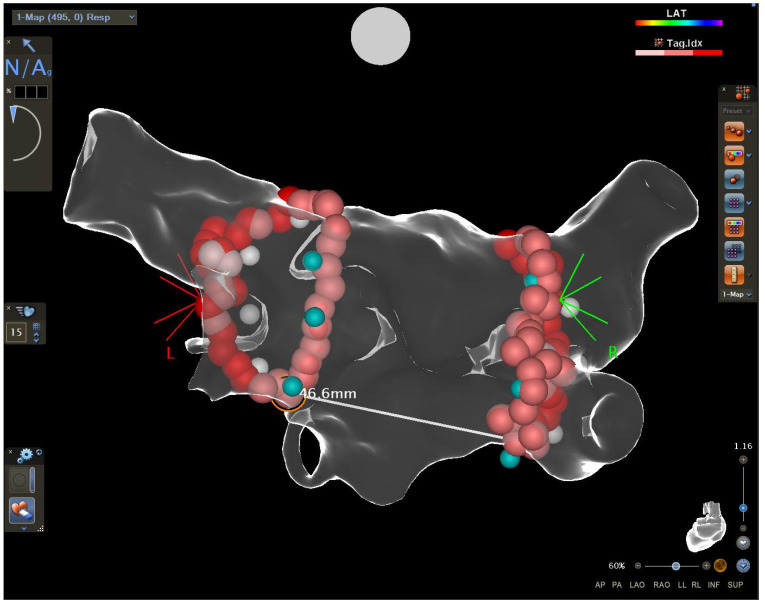
The measurement of the lower distance between the lowest points.

**Figure 3 ijerph-20-05250-f003:**
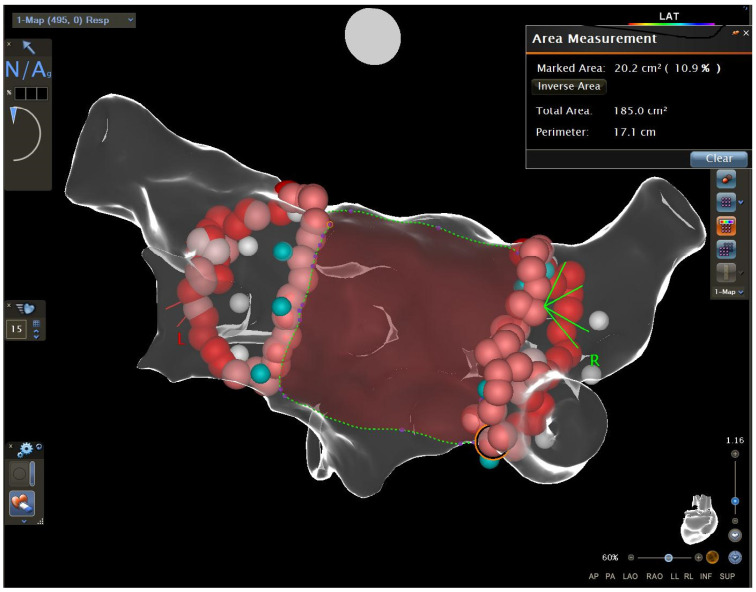
The measurement of the area between the encircling lines on the posterior wall.

**Table 1 ijerph-20-05250-t001:** Demographic and clinical characteristics.

	Study Group (n = 116)
Age (years)	61 ± 10
Male gender, n (%)	74 (64)
BMI (kg/m^2^)	29.7 ± 4.2
Paroxysmal AF, n (%)	77 (66)
Time from AF diagnosis to PVI (years)	3.3 ± 3.0
Arterial hypertension, n (%)	84 (72)
Coronary artery disease, n (%)	23 (20)
Heart failure, n (%)	11 (10)
Diabetes mellitus, n (%)	26 (22)
Left atrial diameter (cm)	4.2 ± 0.5
Left atrial area (cm^2^)	25.8 ± 5.1
Ejection fraction (%)	58.6 ± 7.9
Procedure time (min)	137 ± 26
RF time (min)	30 ± 6

BMI: body mass index; AF: atrial fibrillation; PVI: pulmonary vein isolation; RF: radiofrequency.

**Table 2 ijerph-20-05250-t002:** Analysis of the impact of the distance between lines on first-pass isolation rate.

	First-Pass Isolation Group (n = 68)	Additional Applications Group (n = 48)	*p*-Value
The upper distance between lines (mm)	34.6 ± 8.3	35.1 ± 8.9	0.55
The lower distance between lines (mm)	47.3 ± 8.4	50.8 ± 12.2	0.069
The area between the lines (cm^2^)	12.8 ± 3.4	14.2 ± 5.1	0.070
Area between the lines indexed to left atrial diameter	3.1 ± 0.6	3.0 ± 1.0	0.62
Area between the lines indexed to left atrial area	0.53 ± 0.12	0.54 ± 0.14	0.83

**Table 3 ijerph-20-05250-t003:** Analysis of the impact of the distance between lines on arrhythmia recurrence rate.

	Free from Atrial Arrhythmia Group (n = 85)	Atrial Arrhythmia Recurrences Group (n = 31)	*p*-Value
The upper distance between lines (mm)	35.0 ± 8.6	34.3 ± 8.2	0.71
The lower distance between lines (mm)	48.3 ± 10.2	49.3 ± 9.5	0.64
The area between the lines (cm^2^)	13.2 ± 4.0	13.5 ± 4.3	0.72
Area between the lines indexed to left atrial diameter	3.1 ± 0.8	3.0 ± 0.8	0.52
Area between the lines indexed to left atrial area	0.54 ± 0.13	0.51 ± 0.11	0.48

**Table 4 ijerph-20-05250-t004:** Analysis of the impact of clinical factors on the arrhythmia recurrence rate.

	Free from Atrial Arrhythmia Group (n = 85)	Atrial Arrhythmia Recurrences Group (n = 31)	*p*-Value
Age (years)	61 ± 10	61 ± 10	0.76
Male gender (%)	60	74	0.54
BMI (kg/m^2^)	29.9 ± 4.1	29.6 ± 4.2	0.88
Paroxysmal AF (%)	69	58	0.40
Arterial hypertension (%)	72	74	0.95
Coronary artery disease (%)	16	32	0.066
Heart failure (%)	7	13	0.06
Diabetes mellitus (%)	22	22	0.93
Left atrial diameter (cm)	4.2 ± 0.5	4.3 ± 0.5	0.55
Ejection fraction (%)	59.5 ± 7.0	55.5 ± 10.1	0.014

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
