# Peer review of "The Importance of a Distance between the Lines Encircling Pulmonary Veins in Atrial Fibrillation Ablation on First-Pass Isolation Ratio and Clinical Outcomes"

_ijerph, 2023, doi:10.3390/ijerph20075250_

Round 1

Reviewer 1 Report (New Reviewer)

- The novelties should be highlighted and explained in more detail in the introduction.

- The background section should be improved by adding and reviewing state-of-the-art methods. You can check and cite these papers:

-- Graph-based relevancy-redundancy gene selection method for cancer diagnosis

-- Dual regularized unsupervised feature selection based on matrix factorization and minimum redundancy with application in gene selection

- The three figures of Figures 1-3 can be merged into one Figure.

- Your plan for future work should be explained after the Limitation subsection.

Author Response

The first review (in the second round) seems not to refer to our manuscript. The reviewer asked to cite the publications that do not apply to the subject of our study.

Reviewer 2 Report (New Reviewer)

ijerph-2206831-peer-review-v1

 The importance of a distance between the lines encircling pul-2 monary veins in atrial fibrillation ablation on first-pass isola-3 tion ratio and clinical outcomes by

 Krzyżanowski et al.

This study reviews the data of the authors on the degree of how wide the encircling line is made may influence the out-comes of pulmonary vein isolation (PVI). They hypothesised that the distance between the lines encircling pulmonary veins may correspond with the extent of wide antral circumferential ablation (WACA). The aim of the study was to assess the impact of the distance and the area between the lines on the posterior wall of the left atrium on first-pass isolation rate and 12-13 month freedom from atrial arrhythmia in patients undergoing PVI ablation. Methods and results: 116 patients underwent circumferential ablation index (AI)-guided PVI.

The conclusion is that the distance between the ablation lines did not influence the rate of first-pass isolation and arrhythmia recurrence in the long-term follow-up after PVI procedures.

This is a study with significance in interventional cardiology.  The study design, methods, and results are very specific for interventional cardiology. In addition, the results are negative (reject the hypothesis).  On the basis of the above considerations, this study is more suitable for a specific interventional cardiology journal.

Author Response

The second review (in the second round) suggests that we should sent the manuscript to the journal of specific interventional cardiology. In our opinion the research we did does not only tackle the technical aspects of the atrial fibrillation ablation. The data we shown is certainly important fo interventional cardiologists/electrophysiologists. It can prompt them to change or analyse their own workflow. However, the subject we investigated is also important for general cardiologists and physicians. The distance between the ablation lines is crucial to understanding the general concept of atrial fibrillation ablation. The width of ablation may be of interest for general physician as it shows where the arrhythmogenic foci, arrhythmia triggers and autonomic nervous system of the heart can be. 

This manuscript is a resubmission of an earlier submission. The following is a list of the peer review reports and author responses from that submission.

Round 1

Reviewer 1 Report

In this study, Krzyżanowski et al describe the relationship between PVI lines (as raw distance or ratio) and first pass isolation time/ freedom from any arrhythmia  after 12 months follow up. The question is crucial and could lead to practical changes according to the conclusions

However, modifications have to be done before publications

Minor comments: Follow up is not stringent and results could be affect by this. It has to be mentioned in the limitations

Major comments:

-Here, an heterogeneous population is treated by PVI and we know that PVI is not enough for persistent AF, specially for long standing persistent (50 % efficacy in Star AF2...). Moreover, some RCT studies suggest today to perform ablation beyond PVs (ERASE AF)

To better understand the initial question, it would be preferable to study only paroxysmal AF, where PVI has be shown to be only ablation set performed by the EP community, in first line.

The number of patients will then probably have to be increased

Results and methods have to be more detailed: It would be interesting to have the range and the variability of the area surface, according to the results (and atrial diameters). If the same ablation set is performed with no variability, conclusions would be difficult to interpret 

Author Response

We would like to thank the reviewers for their comments. We will try to correct the manuscript according to the suggestions. Detailed reply is appended below.

Minor comment

The follow-up method was clarified in the revised manuscript. The limitations of the follow-up was mentioned in the revised manuscript.

Major comments

The heterogeneity of the group - paroxysmal, persistent and long-term persistent atrial fibrillation - was mentioned in the limitations of the revised manuscript. However, the degree of WACA depth is important in both paroxysmal and non-paroxysmal AF. The degree of WACA depth in PVI ablation was investigated in few studies. Kiuchi K. et al. examined the importance of the size of the isolated area of the left atrium after PVI for paroxysmal atrial fibrillation [ref 4 in the manuscript]. However, Yoshimura S et al. showed that the outcome of PVI ablation in patient with persistent AF was better when the non-isolated posterior wall area was smaller. Therefore, studies evaluating this aspect of PVI ablation are required not only in paroxysmal but also non-paroxysmal AF.

The fact that in patients with persistent and long-term persistent AF the PVI only was performed in the present study was mentioned in the limitations of the revised manuscript and the appropriate reference was added. However, the trials evaluating the substrate ablation strategies in patients with non-paroxysmal AF yield conflicting results. Furthermore, in 2017-2019, when the procedures in the present study were performed, the evidence to ablate beyond the pulmonary vein isolation was even more scarce.

The range of the distances was added in the revised manuscript according to the reviewer’s suggestion. The variability of the distances and the area of the non-isolated posterior wall seem considerable. Further studies are required to address this aspect of PVI ablation.

Reviewer 2 Report

The Krystian et al revealed that the distance between the ablation line did not influence the rate of first pass isolation and arrhythmia recurrence in the long-term follow-up after PVI procedures incorporating the AI index.

However, I have several concerns.

<major comments>

1,   I think the success rate of PVI depends on risk factor such as obesity, diabetes mellitus, and heart failure. In addition, the kind of AF (paroxysmal, persistent, and long standing) is off course correlated with the recurrence after procedure. Because I think the longer AF persist, the more non-PV foci increase and larger left atrium become.

So authors should evaluate the relationship between the distance of PV lines and recurrence of AF using multivariate analysis involving above factors which are considered to be correlated with AF.

Otherwise, author can not discuss about the conclusion.

At least, author should describe above factors in Table 2.

2, Author should elaborate the incidence of complication in both of group. Readers can not decide whether they make long distance ablation lines or short ablation lines on their own procedure without the information of the rate of complication.

3, How many people receive seven-days Holter monitoring after procedure? What examination was done at follow-up visits? Authors should describe the detail method of follow up visits.    

<minor comment>

Do author ablate posterior wall near esophagus with AI 380?

Author Response

We would like to thank the reviewers for their comments. We will try to correct the manuscript according to the suggestions. Detailed reply is appended below.

Major comments

1. The reason we did not use the multivariate analysis was that the distances between the ablation lines were evenly distributed between the groups with and without recurrences. There was not even the nonsignificant trend. In our opinion the variables included in the multivariate analysis should be at least close to the significancy in univariate analysis (unless there is a known theoretically strong confounder). However, we evaluated the impact of clinical factors on the recurrence rate in the additional table (4) in the revised manuscript according to the reviewer’s suggestion.

2. Complications rate was addressed in the manuscript according to the reviewer’s suggestion. The complications were associated with vascular access and they did not matter in the analysis of the distance between the ablation lines.

3. The follow-up method was clarified in the revised manuscript according to the reviewer’s suggestion.

Minor comment

We ablate at the posterior wall with an AI target 380 (in the past - 2015-2019) and 400 (now). We do not use the intra-oesophageal temperature measurement. We stop the RF delivery at the posterior wall at the AI > 300 in case of severe pain (it was clarified in the revised manuscript).

Round 2

Reviewer 1 Report

Modifications and revisions made lead to a more coherent version

Reviewer 2 Report

I can not understand the P-value about EF on Table 4. It is written as 14, but the author write "a two-side p value <0.05 was considered statistically significant" at statistical analysis section. 

Author Response

Thank you for the suggestions. The number in the Table 4 should be "0.014". It was a typographic error.